# Adaptation of a school-based mental health program for adolescents in Vietnam

**Nga Linh La**[1]*, **Ian Shochet**[2], **Thach Tran**[3], **Jane Fisher**[3], **Astrid Wurfl**[2], **Nga Nguyen**[1], **Jayne Orr**[2], **Ruby Stocker**[3], **Huong Nguyen**[1]

**1** Hanoi University of Public Health, Tu Liem district, Hanoi, Vietnam, **2** School of Psychology and Counselling, Queensland University of Technology, Kelvin Grove, Queensland, Australia, **3** Global and Women's Health, Public Health and Preventive Medicine, Monash University, Melbourne, Victoria, Australia

* aoutlinhnga@gmail.com

## Abstract

### Introduction

Cultural adaptation of a school-based mental health intervention developed in a high-income country is a cost-effective method to address the mental health needs of adolescents in resource-constrained settings. The aim of this study was to translate and culturally adapt the Resourceful Adolescent Program for Adolescents (RAP-A) for adolescents attending high school in Vietnam.

### Methods

The translation and adaptation were conducted using a five-step process including (1) initial stakeholder consultation, (2) forward translation, (3) backward translation, (4) adaptation, and (5) finalising the adapted version. An adaptation panel was established, including the RAP-A authors and mental health and public health experts from Australia, and psychology and public health experts from Vietnam. The panel collaborated closely with a group of stakeholders, including bilingual psychologists and psychiatrists, high school (grades 10–12) students and teachers throughout the adaptation process.

### Results

The adapted version of RAP-A was named 'Happy House'. Happy House was adapted to be delivered in larger groups and in longer sessions than the RAP-A. The 11 sessions in RAP-A were restructured to 6 sessions in Happy House. Major changes were not required for any of the materials. However, some content, illustrations and videos were adapted to be more feasible for the school context and to enhance the comprehensibility, acceptability and appropriateness.

### Conclusion

Happy House has great potential to be relevant, comprehensible and acceptable for Vietnamese adolescents. Further research is warranted to examine the relevance,

**Data Availability Statement:** The interview and group discussion transcripts are available to other researchers upon request, due to ethical restrictions on data sharing. Data contain potentially identifying information. The Information

Sheet given to participants included the following statement: "Recordings and transcripts will be stored in a password-protected folder on a secure drive, accessible only to the named researchers." The Institutional Review Board of the Hanoi University of Public Health has provided the following contact information for any researcher seeking access to our transcripts: Professor Ha Van Nhu, 1A Đức Thắng, Đông Ngặc, Bắc Từ Liêm, Hà Nội, Vietnam, Telephone: + 84 246266 2386 Email: hvn@huph.edu.vn.

**Funding:** This work was supported by the Australian National Health and Medical Research Council [GNT1158429]; the Vietnam National Foundation for Science and Technology Development [NHMRC.108.01-2018.02]. TT is supported by a Monash Strategic Bridging Fellowship; JF is supported by a Finkel Professorial Fellowship, which is funded by Finkel Foundation. The funders had no role in study design, data collection and analysis, decision to publish, or preparation of the manuscript.

**Competing interests:** The authors have declared that no competing interests exist.

comprehensibility, acceptability, and effectiveness of this program on adolescents' mental health before advocating for scaling up program delivery in high schools throughout Vietnam.

## Introduction

Adolescence is a transitional period during which mental health problems have the potential to arise, and if unrecognised, can lead to poor mental health in adulthood [1–3]. Globally, it is estimated that the prevalence of common mental disorders, including depression and anxiety disorders, among adolescents aged 10–19 is 20% [4]. Depression and anxiety disorders are the second and the fourth leading causes, respectively, of years lost due to disability in adolescents aged 15–19 [5], with episodes of depression (subclinical or clinical) in adolescence associated with an increased risk of recurrent depressive episodes in adulthood [6].

Common mental disorders among adolescents are determined by multiple factors. Some factors are shared across the life course, such as poverty, social isolation and loneliness, family conflict, childhood physical and emotional abuse, limited (and/or no) access to mental health services, low social support, low self-esteem, being female, negative cognitive style, and ineffective coping [7–9]. Additionally, there are some factors that are more common during adolescence than in other periods of life, including academic pressure, peer violence and conflict, greater parental control, conflict with parents, and negative body image [9,10].

Schools are an ideal setting for integrating programs that promote physical and mental health among adolescents. In high-income countries, evidence suggests that strategies to integrate mental health interventions into schools, and to use pre-existing resources within schools, are sustainable and effective [11–13]. These kinds of interventions can assist students to build skills in management of emotions and stress, effective communication, and increase students' sense of connection to their school [11,14]. However, there is a notable lack of these programs for children and adolescents in low- and middle-income countries (LMICs) [15–18].

Adapting mental health interventions that have been well established in high-income settings can be a cost-effective method to address and prevent mental health problems in LMICs. The effectiveness of a number of adapted psychological treatments and interventions has been demonstrated in new settings [19,20]. However, to use an evidence-based mental health intervention in a new setting, a rigorous adaptation process, and testing of the adapted program, are always required.

Adaptation is a process that examines the language and contextual aspects of an intervention in a manner that is compatible with the new stakeholders' and users' cultural patterns, meaning and values, to enhance the relevance, acceptability and comprehensibility [21–23]. When an intervention is adapted, it is expected that it will have a better fit to the cultural context, and lead to improved health outcomes [24,25].

There are a variety of models for the adaptation process. The ecological validity model is thought to be the earliest description of an adaptation framework [26]. Bernal et al. [26] proposes eight dimensions of an intervention: language, persons, metaphors, content, concepts, goals, methods, and context, all of which have culturally sensitive elements that require careful development. A cultural accommodation model (CAM) was developed by Leong and Lee [27], extending Leong's integrative model of cross-cultural psychotherapy. This model is comprised of three steps: (1) identify cultural gaps in a theory or intervention that limit its cultural validity; (2) select culturally specific concepts from other culturally diverse psychology to fill gaps;

and (3) test the CAM to determine if its validity is improved compared to the unaccommodated model. Rathod et al. [28] developed a framework for adapting cognitive behaviour therapy (CBT) which involves four stages: (1) review of previous literature and consultation with field experts, as well as gathering of views and experiences from other stakeholders using qualitative methods; (2) emerging themes are used to produce guidance on adapting the CBT manual; (3) translation and adaptation of the materials; and (4) field testing of the adapted materials in a randomised control trial (RCT), followed by further refinement of guidelines.

The community-based participatory research (CBPR) approach is a 'systematic inquiry, with the participation of those affected by the issue being studied, for the purposes of education and taking action or affecting social change' [29]. This approach can achieve trust between researchers and communities; increase the quantity and quality of data; contribute to the emergence of new research questions; and assist the translation of research into locally relevant policies and/or actions [30]. The principles of CBPR are recommended to be included in the adaptation process [31]. Chen et al. proposed the Method for Program Adaptation through Community Engagement, that includes CBPR [32]. An adaption process that involves a partnership between the research team and community members can facilitate shared decision-making and identification of the feasibility and salient issues of the content and delivery methods of the program effectively [33].

The Resourceful Adolescent Program (RAP), a universal school-based program developed in Australia by Shochet and colleagues [34], aims to build psychological resilience and promote positive mental health among young people. The RAP combines elements of cognitive behaviour therapy (CBT) and interpersonal psychotherapy (IPT), which are two interventions with the strongest evidence for targeting adolescent depression [35]. The components of RAP combine to promote self and affect regulation and improve adolescent relationships at the individual, family and school levels. The component of RAP that has been designed for adolescents (RAP-A) is brief and delivered over 11 group-based sessions. Each session is approximately 45 minutes in length and can be run by teachers, psychologists, social workers or other mental health professionals [34]. The main topics of the program are the recognition and affirmation of existing strengths and resources, promoting self-management and self-regulation skills in the face of stress, cognitive restructuring, creating a personal problem-solving model, building and accessing psychological support networks, considering the other's perspective, and keeping and making the peace.

Several RCTs have been conducted to assess the efficiency and effectiveness of RAP. In an Australian trial, when RAP-A was compared to RAP-F (RAP-A plus a program for parents) and Adolescent Watch (normal curriculum), participants in both the RAP-A and RAP-F groups had a significantly greater decrease in depression symptoms post-intervention and at 10-month follow-up, compared to the Adolescent Watch participants [36]. Another trial compared the New Zealand version of RAP to a placebo (no CBT), and found that participants in the intervention group had a significantly greater decrease in two separate depression scores post-intervention, and in one of the scores at 18-month follow-up [37]. Similarly, RAP has been trialled in Mauritius, and compared with a waitlist group [38]. Participants in the intervention group had significant improvements in self-esteem and coping skills at post-intervention and six-month follow-up, as well as lower levels of depression symptoms at post-intervention.

In Vietnam, mental health problems among adolescents are under-recognised. The prevalence of clinically significant symptoms of depression and anxiety among Vietnamese adolescents (up to 41.1% and 22.8%, respectively [39]) are estimated to be among the highest in the world. It is suggested that the education system should take on the role of addressing the needs of children and adolescents, to achieve better mental health in this population [18]. However,

there is a lack of evidence-based, comprehensive, universal mental health programs that can be integrated in schools in Vietnam.

Adopting the framework of adaptation described above, this study aimed to translate and adapt the RAP-A for use in schools in Vietnam.

## Materials and methods

This study was part of a large research project about the RAP-A conducted in Vietnam from May 2019 to June 2021 and included three phases: (1) translation and adaptation, (2) pilot testing, and (3) evaluation of effectiveness. This manuscript reports only the results of Phase One. The results of the other phases will be reported elsewhere.

### Translation and adaptation of RAP-A

There are many program adaptation models that have been developed and used in research settings. However, each model has its own pros and cons and none are translatable across all situations. The adaptation of RAP-A in this study is complex because it includes the translation and adaptation of a psychoeducation program for adolescents in the school context. Therefore, we have selected Rathod et al.' Framework for Adapting Cognitive Behaviour Therapy [28] which is most appropriate to adapt and develop a new model of RAP-A for this study. The model comprised five steps: (1) Initial stakeholder consultation and preparation; (2) Forward translation (3); Back translation into English (4) Adaptation; and (5) Finalising the adapted version. Eight dimensions of an intervention proposed in the Bernal et al.'s framework for ecological validity [26] were examined in every steps of the model. A partnership between the research team and community members (mental health practitioners, adolescents and teachers) was established to facilitate shared decision-making and identification of the feasibility and salient issues of the content and delivery methods of the program as suggested by Chen et al.'s the Method for Program Adaptation through Community Engagement [32].

**Step 1: Initial stakeholder consultation and preparation.** The aims of this step were to obtain the general opinions from different stakeholders of the relevance and acceptability of RAP principles for use in schools in Vietnam, establish the adaptation panel, and finalise the details of the translation and adaptation processes. The adaptation panel consisted of the RAP authors from the Queensland University of Technology (QUT), mental health and public health experts from Monash University (MU), and local psychology and public health experts from the Hanoi University of Public Health (HUPH). The panel purposively selected three groups: (1) local bilingual adolescent mental health practitioners (two psychologists and one psychiatrist); (2) three grade 10 students, and (3) two high school teachers (in Vietnam, high school includes grades 10 to 12) for the initial consultation. The consultation was conducted as a group discussion, and included some panel members and all members of each stakeholder group. From the results of the consultation, the panel made the final decision about the RAP-A materials that would be translated and adapted for use in Vietnam.

**Step 2: Forward translation.** The forward translation of the English version of the selected RAP-A materials into Vietnamese was conducted independently by two bilingual Vietnamese psychologists. All discrepancies were discussed with the three translators and the panel to obtain consensus. The forward translated version formed the first Vietnamese version of RAP-A.

**Step 3: Back translation into English.** This step was conducted to ensure that the translated version included the same content as the original version [40]. The back translation of the

materials was carried out by three other professional local translators who had not been involved in the project and had not read the original version of the materials. The translators worked independently first, and then discussed discrepancies to obtain consensus. The RAP authors checked the back-translated version to identify any content that required specific verification. The adaptation panel then met with the six translators (from Steps 2 and 3) to make any necessary corrections to the Vietnamese version, in order to obtain the second Vietnamese version of RAP-A.

**Step 4: Adaptation.** The aim of this step was to verify the content of the materials and adapt them if required. The three stakeholder groups from Step 1 were included in this step as well. The Vietnamese version 2 of the materials was provided to the stakeholders for review beforehand. Each of the stakeholder groups met with the panel separately, to discuss their suggestions for adaptation. Using results of the discussions with stakeholders, the panel adapted some of the RAP-A content. Local graphic designers and video makers were recruited to assist with adaptation of the visual content.

All of the adaptations were reviewed by the mental health experts, students and teachers to obtain comments and further suggestions. The panel deliberated over the reviewers' comments and suggestions and made necessary changes. This process was repeated until no further issues or suggestions were raised by the stakeholders. The result of this step was the third Vietnamese version of RAP-A.

**Step 5: Finalising the adapted version.** Lastly, all of the adapted content was back-translated into English. The panel reviewed the back-translated content, and finalised the fourth and final Vietnamese version of RAP-A.

**Participants.**   A group of 12 volunteer teachers and 12 Vietnamese researchers attended the facilitator training course. The volunteer teachers were from four urban and rural high schools in Hanoi. The researchers were from HUPH, had experience in youth mental health and psychology, and were members of the Vietnamese Association for Child and Adolescent Psychiatry and Allied Professions (VNACAPAP).

**Data collection.**   The guide for the focus group discussions was developed by the panel. The discussion questions focused on the main themes: relevance of the program, comprehensibility and acceptability of key concepts, metaphors, language and visual content. The focus group discussions were led by two members of the panel from HUPH, were audio recorded, and key notes were made during the focus group discussions.

**Data processing and analysis.**   The data were analysed using content analysis. The focus group discussion transcripts and post-training evaluation form responses were checked and coded by NL and HN, independently. The major themes were relevance; comprehensibility; and acceptability. Afterwards, NL and HN consulted the research team to reach consensus on uncertainties or discrepancies.

**Ethical statement.**   Approval to conduct the study was provided by Monash University Human Research Ethics Committee (Certificate Number: 21455); the Institutional Review Board of the Hanoi University of Public Health (488/2019/YTCC-HD3), Hanoi, Vietnam; and Queensland University of Technology's Office of Research Ethics and Integrity (2000000087).

All activities performed in studies involving human participants were in accordance with the ethical standards of the institutional research committees and with the 1964 Helsinki declaration and its later amendments or comparable ethical standards. Every participant provided written informed consent to participate in this study.

## Results

The adaptation panel was established and consisted of three experts from QUT, including the RAP authors (IS and AW), three mental health and public health experts from MU, and three psychology and public health experts from HUPH.

After the initial stakeholder consultation, the panel decided to translate and adapt all materials of RAP-A [34], comprising the Group Leader's Manual (GLM), the RAP-A Participant Workbook, and video vignettes, to create a Vietnamese version of RAP-A. The GLM is the main guide for facilitators and presents detailed information about the aims and content of each RAP-A session and comprehensive instructions on how to conduct each activity. The GLM is used alongside the RAP-A Participant Workbook. The RAP-A Participant Workbook is provided to each participant and includes all the hand-outs required to complete the program. Throughout the translation and adaptation process, several aspects of RAP-A that needed to be changed or adapted were identified. To summarise, the name of the program, the delivery methods (group size, targeted age group, facilitators, and structure of the program), key words, and some content, illustrations and videos have been adapted to be more culturally appropriate (Table 1).

### Program name

The names 'RAP' and 'RAP-V' have no meaning in Vietnamese. After discussion amongst the panel, we decided to name the Vietnamese version 'Happy House'. This name implies a happy family, comprised of the immediate and extended family members and close friends, and has this specific cultural meaning in Vietnam. Also, this name reflects the main metaphor that runs throughout RAP-A, the 'RAP house', which is derived from *The Three Little Pigs* story. In the story, the 'resourceful little pig' built his house with bricks so that it was strong and resilient; a happy house for the whole family. The family has an essential position in Asian cultures, and in Vietnamese culture in particular.

### Delivery methods

*Group participants and size*. RAP-A is delivered as a group-based program, with approximately 15 adolescents aged 12–15 and a facilitator (group leader). In Vietnam, this age group is split across two separate school levels: secondary (grades 6 to 9: aged 11–15) and high school

**Table 1. Summary of the changes in Happy House.**

| Area | RAP-A | Happy House |
|---|---|---|
| Program name | RAP-A | Happy House |
| Group size | 8 to16 adolescents | 40 to 45 adolescents |
| Age Group | 11–15 | 15–16 |
| Facilitators | 1 | 2 |
| Structure of program | 11x 45-minute weekly sessions | 6x 90-minute weekly sessions |
| Language | English | Vietnamese |
| Content of materials | Include activities that are relevant to adolescents in Australia | Replaced some activities with others that are more appropriate for high school students in Vietnam |
| Illustrations | Images that are relevant to adolescents in Australia. | Some images were redrawn to be more relevant to the high school students in Vietnam |
| Videos | Using Australian characters and activities which are relevant to Australian adolescents. | Videos were remade with Vietnamese characters and activities that are more relevant to high school students in Vietnam |

(grades 10 to 12: aged 15–18). The panel decided to select the older group (high school students) for this program because they usually face more academic pressure than the younger students. If this program is provided to grade 10 students (aged 15–16), it can prepare students to cope with the challenges in their remaining school years.

Grade 10 students are organised into core classes for all subjects. The class sizes range from 40 to 45 students. It is not feasible to divide each class into smaller groups for Happy House because of limitations of the facility and human resources, and the school curriculum. To address this issue, the panel opted to adapt Happy House so that it could be delivered in larger, whole-class groups. This will increase the acceptance when integrating this program into the current high school curriculum.

*Facilitators*. To adjust for the larger group size, the panel suggested that each class should have two facilitators. Group facilitators are school teachers who have completed the mandatory Happy House facilitator training course, and were paired with members of the Vietnamese research team for the pilot.

*Structure of program*. RAP-A is delivered in eleven 45-minute sessions across 11 weeks. Consultations with stakeholders (students and teachers) in Step 1 of the adaptation process revealed that most high school students in Vietnam have a busy schedule and are under significant academic pressure. The stakeholders agreed that if this program was to be delivered over a three-month period, it may be difficult to organise and fit alongside the tight schedule of teaching and examinations. Therefore, Happy House was restructured to fit into six 90-minute sessions (each with a 15-minute break) across six consecutive weeks (Table 2). RAP-A has previously been successfully implemented in fewer than eleven weeks by running multiple sessions consecutively [34].

## Language

There were several key words in the English version of RAP-A for which we could not find a suitable word in Vietnamese that was comprehensible and/or completely equivalent in meaning. This problem was resolved by using short phrases or sentences to express the appropriate meaning in Vietnamese. Some examples are shown in Table 3.

**Table 2. Structure of RAP-A and the Happy House program.**

| RAP-A | | Happy House | |
|---|---|---|---|
| Session | Topic | Session | Topic |
| 1 | Getting to know you! | 1 | Part 1: Getting to know you!<br>Part 2: Feeling good about yourself! |
| 2 | Building self esteem | 2 | Part 1: Introduction to the HH model<br>Part 2: Keep calm |
| 3 | Introduction to the RAP model | 3 | Part 1: Self talk<br>Part 2: Helpful thinking |
| 4 | Keep calm | 4 | Part 1: Finding solutions to problems<br>Part 2: Identifying and accessing support networks |
| 5 | Self talk | 5 | Part 1: Considering the other person's perspective<br>Part 2: Keeping the peace and making the peace |
| 6 | Thinking resourcefully | 6 | Putting it all together |
| 7 | Finding solutions to problems | | |
| 8 | Identifying and accessing support networks | | |
| 9 | Considering the other person's perspective | | |
| 10 | Keeping the peace and making the peace | | |
| 11 | Putting it all together | | |

**Table 3. Examples of language adaptation.**

| English | Problem | Resolved phrase |
|---|---|---|
| Risky and resourceful | Comprehensibility | Depending on the context in the RAP-A material and session, we replaced with 'helpful and unhelpful' (hữu ích và không hữu ích), combined with body clues, self-talk, behaviour, or feelings |
| Resourceful adolescent | Comprehensibility | Adolescents have the ability to better cope with difficult situations (Vị thành niên có khả năng đương đầu tốt hơn với các tình huống khó khăn) |
| Self-esteem | No equivalent word/term | Feeling good about yourself (cảm nhận tốt về bản thân) |
| 'Selfenometer'–a made up metaphor for a self-esteem metric. | Comprehensibility | 'Feeling good about yourself' Measure (Thước đo cảm nhận tốt về bản thân) |
| Bubble behaviour, thought, feelings | Comprehensibility | Circle behaviour, thought, feelings (Vòng tròn hành vi, suy nghĩ, cảm nhận) ('Circle' is more easily understood than 'bubble' in Vietnamese) |
| 'Thought Court'- a program activity for cognitive restructuring. | Comprehensibility | We will explain: 'It means that we challenge our thoughts and decide whether they are true or false, unhelpful or helpful' |

## Content of the materials

Changes were not required for the majority of the materials. There was some content that the panel decided to adapt, including activities that are not feasible for the school context, and some elements that needed enhanced comprehensibility and appropriateness. The details of the changes are shown in Table 4.

## Illustrations

There were only two images that were identified for adaptation. These two images were amended so that they are more familiar to Vietnamese adolescents (Figs 1 and 2).

## Videos

RAP-A has five short videos to demonstrate some activities. Three videos, 'Saskia', 'Tom Needs a Project Partner' and 'Amanda Gore', needed to be remade using Vietnamese characters. The other two videos are cartoons that could be retained, with Vietnamese subtitles added.

*Saskia Video.* In the new video, we changed the English names (Saskia and Michelle) to Vietnamese names (Mai and Minh). We also replaced the text messages in the original video with Facebook messages, as this is a more common communication method for Vietnamese adolescents. Fig 3 shows the original and Vietnamese actresses in the Saskia video.

*'Tom Needs a Project Partner' Video.* The new video has the same content. The character's name Tom was changed to Phong, a common Vietnamese male name.

*'Amanda Gore' Video.* Amanda Gore is an internationally renowned motivational speaker and author. She is famous in many countries, but is not well known in Vietnam. Therefore, in the new video, Amanda Gore was replaced with a Vietnamese female teacher. The content was kept the same as the original video.

## Discussion

The aim of this study was to translate and adapt RAP-A for adolescents attending high school in Vietnam. This was achieved by following a process of translation and adaptation adapted from several of the most widely-used program adaptation models in health and psychology [26,28,32]. To our knowledge, this study is the first to translate and adapt a universal school-based mental health intervention for Vietnamese adolescents. The adapted version of RAP-A,

**Table 4. Content adaptation.**

| Activity | Problem | Adaptation |
|---|---|---|
| Activity 1b 'This is Andrew and he likes abseiling' | The name Andrew is not familiar in Vietnam. Abseiling is not a common hobby for Vietnamese adolescents | A common Vietnamese name was selected to replace Andrew. Abseiling was replaced with reading. |
| Optional activity: 'Post me a note': Each student writes their name on the bottom of a large piece of paper and passes their paper to the person on their right. The receiving person writes a positive comment about the person whose name appears on the paper, folds the comment down, and then passes it onto the person on their right. | The class size is too large to manage this activity | This activity was removed |
| Optional activity: 'wordles' | English language-based activity, difficult to translate | This activity was removed |
| Role play: "All your friends are going to the movies and mum and dad won't let you go" | 'Going to the movies' is unpopular with Vietnamese students, especially in rural areas. | Changed scenarios: Going to the movies -> going out to eat |
| Role play: "Your dad takes your brother fishing although he has been promising to take you" | Going fishing is an uncommon activity for Vietnamese adolescents | "Go fishing" was changed to "go shopping" |
| Role play: "A group of your friends has arranged to have a sleepover at a friend's house. Your parents don't know their parents, but you would really like to go" | Sleepovers are unpopular with Vietnamese adolescents | The scenario was changed to "You want to go out after school with your friends, but your parents want you to come straight home and do your homework." |
| 'Support network' bricks | Australian networks | We replaced the list of Australian networks with available and appropriate services in Vietnam. |
| Optional activity: resilience kits (Items: cellophane, eraser, Kit Kat, stress ball, paperclip, rubber band, strength card, marbles, etc.) *Every student will be given items to put in their box to make a "resilience kit"* | This activity needs some preparation that would not be easy when scaling up. | This activity was removed |

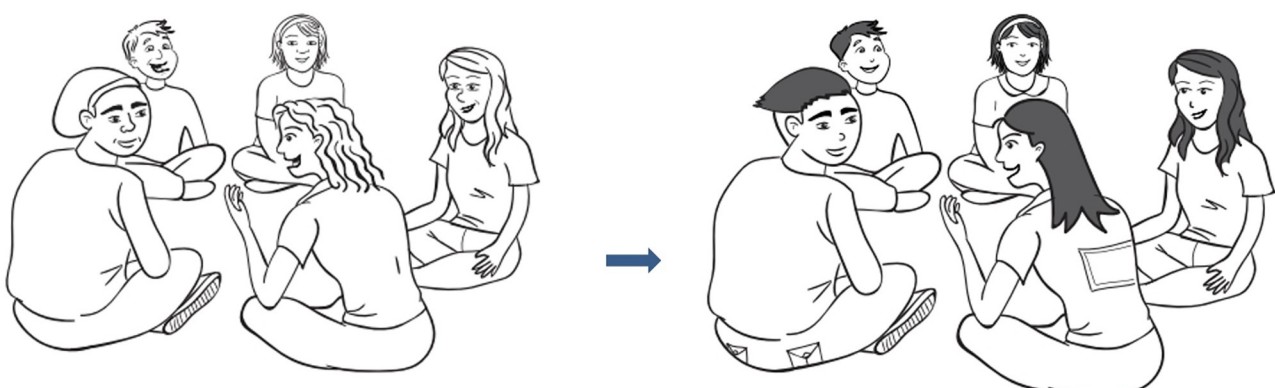

**Fig 1. Original and Vietnamese versions of the Group Rule image.** Reprinted from the RAP-A and Happy House Participant Workbooks under a CC BY license, with permission from Astrid Wurfl, original copyright 2021.

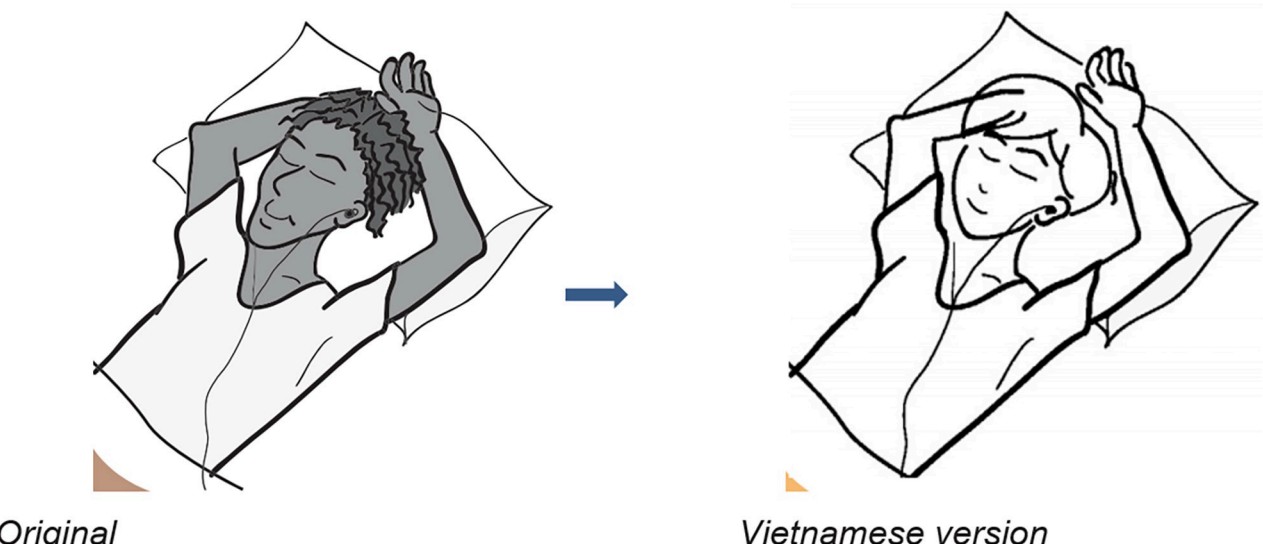

**Fig 2. Original and Vietnamese versions of the Relaxation Brick image.** Reprinted from the RAP-A and Happy House Participant Workbooks under a CC BY license, with permission from Astrid Wurfl, original copyright 2021.

Happy House, has great potential to be highly relevant, comprehensible and acceptable for students in high schools in Vietnam.

There were several important adaptations to the delivery of the program. RAP-A was designed to be implemented in groups of approximately 15 students. Adapting RAP-A to be a universal school-based intervention in Vietnam meant that the group size for Happy House is the size of a class, approximately 45 students. The larger group size can introduce some management challenges for facilitators. We have developed detailed guidelines and carefully adapted the GLM to address these potential challenges. However, future studies are warranted to examine the effectiveness of Happy House with larger groups. Booster sessions for students using online resources, mobile phone applications, social media, or SMS should be considered to increase the effectiveness of the program [31].

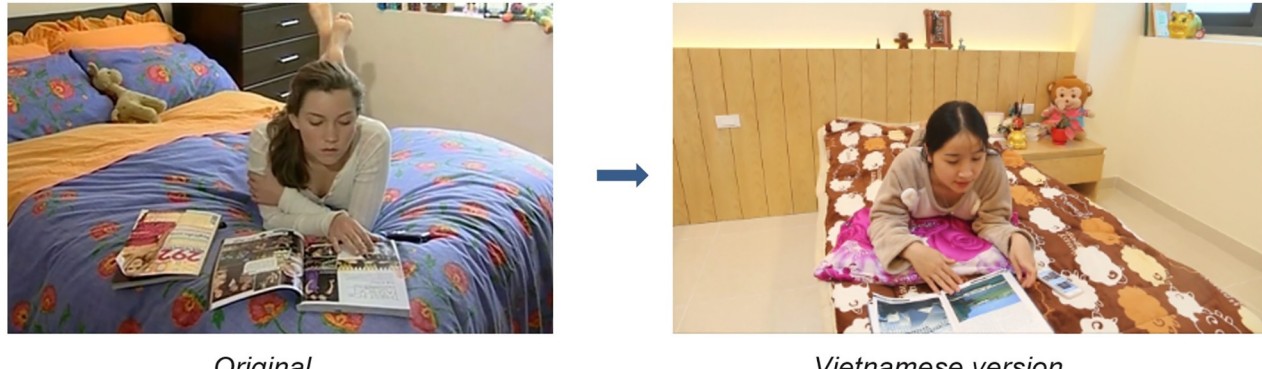

**Fig 3. The original and Vietnamese actresses in the Saskia video.** Reprinted from the RAP-A and Happy House videos under a CC BY license, with permission from Astrid Wurfl, original copyright 2021.

An additional challenge faced in this study was the translation of the key concepts of the program. For instance, we could not find an equivalent word in Vietnamese for 'self-esteem'. In previous health research and documents in Vietnam, 'self-esteem' was translated as 'self-evaluation', 'self-confidence' or 'egotism' [41]. These words do not express the meaning of the concept completely. Furthermore, these Vietnamese translations were found to be difficult for stakeholders to comprehend. The solution we found was to use a phrase; for example, 'self-esteem' was replaced with 'feeling good about yourself'. The results of the pilot tests suggested that this approach was effective and that phrases were understandable.

RAP was originally designed for small groups of adolescents aged 12–15 in an English-speaking, high-income country. This study suggests that the approach and content, after rigorous adaptation, are relevant and acceptable to students in a resource-constrained developing country. The detailed process of translation and adaptation of RAP-A allowed for identification of issues related to language, delivery methods, content, metaphors and illustrations that are beyond the scope of simple translation. This process helped us to understand and elucidate the core factors and details that need to be adapted so that they could be applied in Vietnam. The results demonstrated the value and importance of collaboration between experts and community members in the adaptation process [33]. Community stakeholders (in this study: local experts, students and high school teachers) were involved in most steps of the translation and adaptation process as well as in the pilot testing of the program. The tight collaboration with the stakeholders and shared decision-making from the first phase of the study resulted in an adapted version of the program that was consistently found to be highly relevant, comprehensible and acceptable in the pilot testing. The process we proposed and used in this study offers a model for future translation and adaptation of mental health interventions in Vietnam and other LMIC.

## Limitations

We acknowledge that the main limitation of this study was that parents, a key stakeholder group, were not included in the adaptation panel. Some parents were invited to participate in the panel, but declined.

## Conclusions

The findings of this study indicate that the adapted RAP-A program has great potential to be relevant, comprehensible and acceptable to adolescents attending high school in Vietnam. It also suggests that there is potential to integrate this program into the current high school curriculum in Vietnam. However, the effectiveness of Happy House in impacting adolescents' mental health outcomes needs to be evaluated, before advocating for scaling up the program.

## Acknowledgments

The authors are especially grateful to the mental health practitioners, teachers and adolescents who participated in this research.

## Author Contributions

**Conceptualization:** Nga Linh La, Ian Shochet, Thach Tran, Jane Fisher, Astrid Wurfl, Huong Nguyen.

**Data curation:** Nga Linh La, Nga Nguyen.

**Formal analysis:** Nga Linh La.

**Funding acquisition:** Thach Tran, Jane Fisher, Huong Nguyen.

**Methodology:** Nga Linh La, Thach Tran, Jane Fisher, Huong Nguyen.

**Project administration:** Nga Nguyen.

**Supervision:** Nga Linh La, Ian Shochet.

**Validation:** Nga Linh La.

**Writing – original draft:** Nga Linh La, Thach Tran, Ruby Stocker.

**Writing – review & editing:** Nga Linh La, Ian Shochet, Thach Tran, Jane Fisher, Astrid Wurfl, Nga Nguyen, Jayne Orr, Ruby Stocker, Huong Nguyen.

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
