## [Decision Letter · Decision Letter 0]

21 Apr 2022

PONE-D-22-01625A school-based mental health program for adolescents in Vietnam: adaptation, relevance, comprehensibility and acceptabilityPLOS ONE

Dear Dr. La,

Thank you for submitting your manuscript to PLOS ONE. After careful consideration, we feel that it has merit but does not fully meet PLOS ONE’s publication criteria as it currently stands. Therefore, we invite you to submit a revised version of the manuscript that addresses the points raised during the review process.

We look forward to receiving your revised manuscript.

Kind regards,

Bronwyn Myers

Academic Editor

PLOS ONE

Journal Requirements:

a) Did participants provide their written or verbal informed consent to participate in this study?

3. We note that your paper includes detailed descriptions of individual patients/participants. As per the PLOS ONE policy (http://journals.plos.org/plosone/s/submission-guidelines#loc-human-subjects-research) on papers that include identifying, or potentially identifying, information, the individual(s) or parent(s)/guardian(s) must be informed of the terms of the PLOS open-access (CC-BY) license and provide specific permission for publication of these details under the terms of this license. Please download the Consent Form for Publication in a PLOS Journal (http://journals.plos.org/plosone/s/file?id=8ce6/plos-consent-form-english.pdf). The signed consent form should not be submitted with the manuscript, but should be securely filed in the individual's case notes. Please amend the methods section and ethics statement of the manuscript to explicitly state that the patient/participant has provided consent for publication: “The individual in this manuscript has given written informed consent (as outlined in PLOS consent form) to publish these case details

4. We note that you have stated that you will provide repository information for your data at acceptance. Should your manuscript be accepted for publication, we will hold it until you provide the relevant accession numbers or DOIs necessary to access your data. If you wish to make changes to your Data Availability statement, please describe these changes in your cover letter and we will update your Data Availability statement to reflect the information you provide

5. We note that Figure(s) 1 and 2 in your submission contain copyrighted images. All PLOS content is published under the Creative Commons Attribution License (CC BY 4.0), which means that the manuscript, images, and Supporting Information files will be freely available online, and any third party is permitted to access, download, copy, distribute, and use these materials in any way, even commercially, with proper attribution. For more information, see our copyright guidelines: http://journals.plos.org/plosone/s/licenses-and-copyright.

1. You may seek permission from the original copyright holder of Figure(s) 1 and 2 to publish the content specifically under the CC BY 4.0 license. 

Reviewers' comments:

Reviewer's Responses to Questions

**Comments to the Author**

1. Is the manuscript technically sound, and do the data support the conclusions?

Reviewer #1: Yes

Reviewer #2: Partly

2. Has the statistical analysis been performed appropriately and rigorously? 

Reviewer #1: N/A

Reviewer #2: N/A

3. Have the authors made all data underlying the findings in their manuscript fully available?

Reviewer #1: No

Reviewer #2: No

4. Is the manuscript presented in an intelligible fashion and written in standard English?

Reviewer #1: Yes

Reviewer #2: Yes

5. Review Comments to the Author

Reviewer #1: This manuscript describes the process of translating, adapting, and testing a school-based mental health from a different country RAP-A to "Happy House".

General Comments

I first want to applaud the authors for documenting their adaptations and preparing this manuscript. I firmly believe more papers like this need to be published. In general, I am very much interested in seeing this paper published myself. I would however suggest that the authors work to streamline the changes and make them more easily understandable to the reader. I feel that at this time, the adaptations are mixed in w/other information and it is hard to follow.

Perhaps one way to approach this is a summary paragraph at the beginning of the results section that walks the reader through the various adaptions and then have sections by each of those adaptions area. Or perhaps a table of the changes (similar to Table 1) where you document the components of the RAP-A program and then your adaptions on the left under "Happy House". Something very simplistic for the table such as "Program Name; RAP'A; Happy House"; then "Group size; x; x" and "Age group; 12-15 years; 11-18"; etc. This will help the reader quickly see what changed and the differences. Then your results section can discuss in more details.

Lastly, I feel this is two papers in one. You should have one paper on the changes and the second on the pilot data. Right now having the two in one paper is too much and the details of the pilot get lost. Each of these (the modifications, and the pilot) deserve their own paper.

For the pilot test outcomes paper your table 1 should be the demographics of who the students/teachers were. This way others that may want to use your program know more details about the demographics of your population and then can generalize them (or not) to their community.

Example of places that need flushed out more:

• Page 13, lines 256-7: Authors state "several issues arose"; however, the only problem discussed below is the program name. Can the authors please give more details regarding the "issues"?

Page 17, line 306: "some examples". Again, please be clear what was changed.

Reviewer #2: This is an interesting and important piece of work. The authors attempt to cover a lot of ground by reporting on both the adaptation process and the pilot/feasibility test of the intervention. Because of this, the paper is light on some of the details of the pilot test. I believe that this could be partitioned off to be a second paper- allowing for more detail on the pilot to be provided.

I also found the use of three approaches to cultural adaptation confusing- the rationale for using aspects of these three approaches rather than choosing one approach is not clear enough.

6. PLOS authors have the option to publish the peer review history of their article (what does this mean?). If published, this will include your full peer review and any attached files.

Reviewer #1: No

Reviewer #2: No

---

## [Decision Letter · Decision Letter 1]

12 Jul 2022

Adaptation of a school-based mental health program for adolescents in Vietnam

PONE-D-22-01625R1

Dear Dr. La,

We’re pleased to inform you that your manuscript has been judged scientifically suitable for publication and will be formally accepted for publication once it meets all outstanding technical requirements.

Kind regards,

Bronwyn Myers

Academic Editor

PLOS ONE

Additional Editor Comments (optional):

Reviewers' comments:

Reviewer's Responses to Questions

**Comments to the Author**

1. If the authors have adequately addressed your comments raised in a previous round of review and you feel that this manuscript is now acceptable for publication, you may indicate that here to bypass the “Comments to the Author” section, enter your conflict of interest statement in the “Confidential to Editor” section, and submit your "Accept" recommendation.

Reviewer #1: All comments have been addressed

Reviewer #2: All comments have been addressed

2. Is the manuscript technically sound, and do the data support the conclusions?

Reviewer #1: Yes

Reviewer #2: (No Response)

3. Has the statistical analysis been performed appropriately and rigorously? 

Reviewer #1: Yes

Reviewer #2: (No Response)

4. Have the authors made all data underlying the findings in their manuscript fully available?

Reviewer #1: Yes

Reviewer #2: (No Response)

5. Is the manuscript presented in an intelligible fashion and written in standard English?

Reviewer #1: Yes

Reviewer #2: (No Response)

6. Review Comments to the Author

Reviewer #1: This condensed manuscript reads much better. I like that it is focused on the changes that you made vs the pilot results. This will set up your next paper nicely. Great work!

Reviewer #2: (No Response)

7. PLOS authors have the option to publish the peer review history of their article (what does this mean?). If published, this will include your full peer review and any attached files.

Reviewer #1: No

Reviewer #2: No

---

## [Editor Report · Acceptance letter]

15 Jul 2022

PONE-D-22-01625R1 

Adaptation of a school-based mental health program for adolescents in Vietnam 

Dear Dr. La:

I'm pleased to inform you that your manuscript has been deemed suitable for publication in PLOS ONE. Congratulations! Your manuscript is now with our production department. 

Kind regards, 

on behalf of

Dr. Bronwyn Myers 

Academic Editor

PLOS ONE